# Assessment of Endothelial Injury and Pro-Coagulant Activity Using Circulating Microvesicles in Survivors of Allogeneic Hematopoietic Cell Transplantation

**DOI:** 10.3390/ijms21249768

**Published:** 2020-12-21

**Authors:** Eleni Gavriilaki, Ioanna Sakellari, Panagiota Anyfanti, Ioannis Batsis, Anna Vardi, Zoi Bousiou, Antonios Lazaridis, Barbara Nikolaidou, Ippokratis Zarifis, Marianna Masmanidou, Efthalia Yiannaki, Dimitra Markala, Achilles Anagnostopoulos, Stella Douma, Eugenia Gkaliagkousi

**Affiliations:** 1Hematology Department-BMT Unit, G Papanicolaou Hospital, 57010 Thessaloniki, Greece; ioannamarilena@gmail.com (I.S.); iobats@yahoo.gr (I.B.); anna_vardi@yahoo.com (A.V.); boussiou_z@hotmail.com (Z.B.); mariannareti@gmail.com (M.M.); achanagh@gmail.com (A.A.); 23rd Department of Internal Medicine, Papageorgiou Hospital, Aristotle University of Thessaloniki, 56403 Thessaloniki, Greece; panyfan@hotmail.com (P.A.); spanbiol@hotmail.com (A.L.); barbienn@yahoo.gr (B.N.); zarifis.ipp@gmail.com (I.Z.); sdouma@auth.gr (S.D.); eugalant@yahoo.com (E.G.); 3Hematology Laboratory, Theagenion Cancer Center, 54007 Thessaloniki, Greece; eyiannaki@gmail.com (E.Y.); dimarkala@gmail.com (D.M.)

**Keywords:** allogeneic hematopoietic cell transplantation, vascular injury, pro-coagulant activity, microvesicles

## Abstract

(1) Background: survivors of allogeneic hematopoietic cell transplantation (alloHCT) suffer from morbidity and mortality due to cardiovascular events. We hypothesized that vascular injury and pro-coagulant activity are evident in alloHCT survivors without existing alloHCT complications or relapse. (2) Methods: we enrolled consecutive adult alloHCT survivors without established cardiovascular disease and control individuals matched for traditional cardiovascular risk factors (January–December 2019). Circulating microvesicles (MVs) of different cellular origins (platelet, erythrocyte, and endothelial) were measured by a standardized flow cytometry protocol as novel markers of vascular injury and pro-coagulant activity. (3) Results: we recruited 45 survivors after a median of 2.3 (range 1.1–13.2) years from alloHCT, and 45 controls. The majority of patients suffered from acute (44%) and/or chronic (66%) graft-versus-host disease (GVHD). Although the two groups were matched for traditional cardiovascular risk factors, alloHCT survivors showed significantly increased platelet and erythrocyte MVs compared to controls. Within alloHCT survivors, erythrocyte MVs were significantly increased in patients with a previous history of thrombotic microangiopathy. Interestingly, endothelial MVs were significantly increased only in alloHCT recipients of a myeloablative conditioning. Furthermore, MVs of different origins showed a positive association with each other. (4) Conclusions: endothelial dysfunction and increased thrombotic risk are evident in alloHCT recipients long after alloHCT, independently of traditional cardiovascular risk factors. An apparent synergism of these pathophysiological processes may be strongly involved in the subsequent establishment of cardiovascular disease.

## 1. Introduction

Allogeneic hematopoietic cell transplantation (alloHCT) is the only curative option for various hematologic malignant and non-malignant diseases [1]. Although major advances in the standard of care have occurred [2], cardiovascular (CV) events are recognized as a major cause of morbidity and mortality among alloHCT patients that did not present with relapse or secondary malignancy. Interestingly, CV events are only second to the devastating condition of graft-versus-host disease (GVHD) [3,4]. Subsequently, CV mortality is twice up to 4-fold higher than that of the general population, and increases as time passes from HCT [5]. Despite the renewed interest in the field, recent studies have focused mainly on traditional CV risk factors, such as the metabolic syndrome [6] and hypertension [7]. Little is known about earlier prediction of CV risk that is needed in order to prevent CV morbidity and mortality in these patients [8].

Regarding CV research, subclinical target organ damage has been clearly defined in order to implement CV risk stratification and early treatment interventions [9]. In addition, vascular injury is established as an early event in the pathophysiology of CV risk. Several markers have been suggested for assessment of vascular injury [10]. Microvesicles (MVs) have also been recently acknowledged as potential markers in patients with high CV risk [11,12].

Depending on their origin, MVs reflect the function of the cell they stem from. Therefore, endothelial and platelet MVs have been mainly studied in CV research. Endothelial MVs have emerged as multifaceted biological conveyors at the crossroad of inflammation, thrombosis and angiogenesis [13]. They are characterized by significant procoagulant and proinflammatory properties, directly linked to endothelial dysfunction [14]. On the other hand, platelet MVs have not been considered only markers of platelet activation, but also reflect a potent procoagulant activity [15]. Furthermore, platelet microvesicles (PMVs) participate in several homeostatic responses and pathogenic processes including inflammation, atherosclerosis, immunity, and cancer [16]. Recently, erythrocyte MVs have also been implicated in patients with increased CV risk [12], with emerging evidence from basic research that suggests a potent thrombo-inflammatory role [17]. Such novel markers also aim to better characterize the substantial residual CV risk in high-risk patients, which is not effectively treated with current therapeutic options [18,19]. Interestingly, a recent report states that as many as 20% of patients after an acute coronary syndrome suffer a recurrent event within 12 months despite effective dual antiplatelet treatment [20].

Vascular injury and pro-coagulant activity have been linked to certain acute complications of alloHCT, such as transplant-associated thrombotic microangiopathy [21,22]. However, studies of alloHCT survivors are scarce. Given the lack of previous studies in the field, we hypothesized that vascular injury and procoagulant activity are evident in alloHCT survivors without ongoing alloHCT complications or relapse. To this end, we compared alloHCT recipients with non-HCT individuals, matched for traditional cardiovascular risk factors and measured circulating MVs as robust markers of vascular damage and prothrombotic tendency.

## 2. Results

### 2.1. Study Population

We studied 45 patients after a median of 2.3 (range 1.1–13.2) years from alloHCT, who were free from established CV disease. Patients’ pre-transplant and transplant characteristics are shown in Table 1. It should be noted that the majority of alloHCT recipients had received treatment for acute (44%) and/or chronic GVHD (66%) before enrollment in the study. When studied, alloHCT recipients were not on immunosuppressive treatment and had no signs or symptoms of severe complications, active malignancy or relapse. In addition, alloHCT recipients had full hematopoietic reconstitution. In parallel, we studied 45 non-HCT control individuals, which were matched for traditional CV risk factors (age, hypertension, diabetes, dyslipidemia, obesity, smoking) with alloHCT survivors.

### 2.2. Traditional Markers of Cardiovascular Disease

According to the study design, traditional cardiovascular risk factors did not differ between groups (Table 2). Interestingly, 7 out of 45 patients (16%) had been already diagnosed and treated for hypertension before the study visit (2 in the age group of 18–39 years and 5 in the group of 40–59 years). Furthermore, 4 out of 45 patients (9%) were diagnosed with hypertension based on the measurements of the present study (1 in the age group of 18–39 years and 5 in the group of 40–59 years). In addition, 10 patients suffered from dyslipidemia and 2 from diabetes mellitus.

### 2.3. Novel Markers of Vascular Injury and Increased Thrombotic Risk

Erythrocyte and platelet MVs were significantly increased in alloHCT recipients compared to controls (*p* = 0.004 and *p* = 0.039, respectively). Although endothelial MVs were increased in alloHCT recipients, the difference did not reach statistical significance (*p* = 0.133). There was also a trend towards a decrease of endothelial MVs as the interval after alloHCT increases (*p* = 0.094). Table 3 summarizes circulating levels of MVs in both groups.

We further investigated possible associations of these novel markers in alloHCT recipients. Interestingly, endothelial MVs were significantly increased in the majority of alloHCT recipients that had received a myeloablative conditioning (*n* = 32/45, *p* = 0.012), compared to recipients of non-myeloablative conditioning. In addition, erythrocyte MVs were significantly increased in patients with a history of transplant-associated thrombotic microangiopathy (*n =* 3/45, *p* = 0.021), compared to patients with transplant-associated thrombotic microangiopathy. The latter had been treated as previously described [23,24] and resolved before a median of 13 (11–20 months) in these three patients. It is noteworthy that significant associations were found among MVs of different origins. As expected, platelet MVs were significantly associated with endothelial (r = 0.333, *p* = 0.002) and erythrocyte (r = 0.249, *p* = 0.034) MVs, suggesting an interaction between MVs of different origin

## 3. Discussion

Our findings suggest for the first time that endothelial injury and pro-coagulant activity are evident in adult alloHCT survivors, as this is reflected by the measurement of circulating MVs, independently of traditional CV risk factors. Furthermore, we were able to identify high-risk populations based on clinical features.

Over the last decades, our understanding of vascular injury has substantially improved. It is considered an early event in the pathophysiology of CV disease, contributing to subclinical target organ damage [25,26]. Biomarkers and potential treatment strategies for thrombotic events remain under investigation [27,28,29,30]. In 1992, Celermajer et al. reported the first non-invasive endothelial function test [31]. Since then, a plethora of vascular indices and biomarkers have been suggested for the assessment of endothelial dysfunction [32]. Among them, endothelial MVs are characterized by significant biological properties implicating them in several pathophysiological pathways, including the pathogenesis of CV diseases. Elevated endothelial MVs have been reported in patients with a variety of CV comorbidities, including diabetes, hypertension, acute coronary syndromes, and chronic ischemic heart disease [11,12,33]. Not only do they represent markers of endothelial dysfunction, but are also considered conveyors of biological messages between cells, which are implicated in the evolvement of vascular damage.

In alloHCT survivors, indices of endothelial injury have been evaluated in limited recent studies with heterogenous populations [34,35,36]. Using the gold standard vascular method of flow-mediated dilatation (FMD), autologous and allogeneic HCT recipients had impaired endothelial dysfunction compared to values prior HCT [34] and their siblings [36]. By contrast, when compared to age- and sex-matched controls, no difference in endothelial dysfunction was observed in childhood alloHCT survivors [35]. Regarding biomarkers of endothelial function, endothelial MVs have been scarcely studied in HCT recipients. Endothelial MVs were increased during the early post-transplant period (2–3 weeks post transplantation) [37] and in acute GVHD patients [38]. Nevertheless, none of these studies investigated a link between endothelial dysfunction and CV risk in HCT survivors. In our well-designed study of alloHCT recipients and controls, endothelial MVs were significantly increased only in patients receiving a myeloablative conditioning. Since myeloablative conditioning remains the backbone of alloHCT strategies [39,40], our finding is of great importance, since it is suggestive of sustained underlying endothelial damage, at least in this specific subgroup of patients.

In terms of pro-coagulant activity, this has been mainly studied in patients with acute endothelial syndromes, such as transplant-associated thrombotic microangiopathy [21] or GVHD [41,42]. Indeed, these recent studies have proven a crosstalk between endothelial dysfunction and thrombotic risk during acute syndromes. Nevertheless, this aspect has not been studied in alloHCT survivors, especially with regards to CV risk. In order to investigate if there is prothrombotic tendency in alloHCT survivors, after adjusting for the existence of CV risk factors, we measured promising markers of thrombosis already investigated in the field of CV disease, which is platelet and erythrocyte MVs [12,43]. Indeed, a link with thrombosis has been established not only for platelet MVs, but also recently for erythrocyte MVs [17,44]. Our study confirms for the first time this link in alloHCT recipients, suggesting that vascular injury and pro-coagulant activity may represent mutually reinforcing and interdependent processes in these patient group, independently of the presence of traditional CV risk factors.

Lastly, a really interesting finding of our study is the increase of erythrocyte MVs in patients with a history of transplant-associated thrombotic microangiopathy. This finding implicates erythrocyte MVs a potential long-lasting markers of thrombo-inflammation that is evident in these patients. Because the sample size was too small (only three patients), this result should mainly serve as an indication pointing towards future research. However, the aforementioned condition is rare and the patient’s survival from this condition even rarer [45,46]. In this sense we reckon that this finding should be highlighted keeping in mind of course the above limitation.

The present study has some limitations and strengths. Firstly, the observational nature of the study does not allow for causality assumptions or for serial measurements of MVs before and after alloHCT. Since there were no similar studies in the field, the relatively small sample size was based on sample size calculations. Our group of alloHCT survivors could be considered heterogeneous regarding the disease type and phase, as well as the timing post alloHCT. It should be also noted that our study cannot assess the effect of the patients’ disease itself or pre-alloHCT treatments on endothelial injury and pro-coagulant activity. Nevertheless, it is a meticulously selected population of alloHCT survivors and matched controls that provides novel data on vascular injury and pro-coagulant activity in alloHCT survivors, highlighting the clinical characteristics of high-risk patients. Further prospective studies are needed to address the role of novel markers in CV morbidity and mortality of alloHCT survivors, as well as the effects of disease type, phase, and pre-transplant treatments on these markers.

## 4. Materials and Methods

### 4.1. Study Population

We enrolled consecutive adult alloHCT survivors from our HCT Clinic (January–December 2019), with a follow-up from alloHCT longer than one year. We excluded patients with established cardiovascular disease, defined as stroke, angina, ischemic heart disease, heart failure, and arrhythmias; chronic graft-versus-host disease under immunosuppressive treatment (GVHD), acute or chronic inflammatory disease, active malignancy or relapse. All patients underwent allogeneic HCT at our JACIE (Joint Accreditation Committee-ISCT and EBMT) accredited Unit. Data relevant to alloHCT and post-HCT follow-up were collected retrospectively from our prospectively acquired database. AlloHCT was performed according to our standard operational procedures, as previously published [39,40,47,48].

We also enrolled consecutive control individuals from our Outpatient Hypertension Clinic who attended regular appointments, matched for traditional cardiovascular risk factors (age, hypertension, diabetes, dyslipidemia, obesity, and smoking) with alloHCT survivors. In accordance with the Helsinki Declaration, all patients have given written informed consent. All subjects were of Caucasian origin. The institutional review board of Aristotle University of Thessaloniki approved our study.

### 4.2. Laboratory Measurements

All measurements were performed between 9:00 and 11:00 a.m., with the participants having refrained from food, coffee, and smoking for at least 10 h. Detailed history, physical examination, and blood sampling were performed.

Office blood pressure (BP) was measured in the sitting position using a validated oscillometric device (Microlife Exact BP, Microlife AG, Widnau, Switzerland), according to standard recommendations for office BP measurement [27]. The mean of the second and third value of three consecutive measurements with a 2-min interval in the arm with the higher BP was considered as the patients’ office BP. Hypertension was defined according to current guidelines [49].

#### 4.2.1. Biochemical Profile

Plasma glucose, lipids (total cholesterol, LDL-cholesterol, HDL-cholesterol, triglycerides), renal and liver function were determined using routine laboratory techniques under fasting conditions.

#### 4.2.2. Blood Sampling for MVs Quantitation

Blood samples for MV measurements were collected in citrated tubes (sodium citrate 3.2%) and centrifuged within 30 min with a two-step centrifugation protocol (2500× *g* for 15 min at room temperature, followed by a second centrifugation of the supernatant at the same condition). Supernatant was collected, and platelet poor plasma was stored at −80 °C [29]. MVs detection was then performed on thawed samples using a CyFlow Cube8 ROBBY flow cytometer (Sysmex Partec GmbH, Goerlitz, Germany). Preliminary experiments showed excellent reproducibility in samples stored less than 2 weeks (coefficient of variation <10%). Therefore, all samples were analyzed in less than 2 weeks from collection.

#### 4.2.3. MVs Quantitation

Flow cytometry protocol for MVs quantitation was performed as previously published [11,12]. Background noise was checked using ultrapure water at less than 2000 events/s (flow rate ~10,000 events/s. The following fluorochrome coupled antibodies and their corresponding isotypes were used for MVs detection: anti-CD105 (Cluster of Differentiation 105) and anti-CD144 PE (phycoerythrin) (Immunostep, Salamanca Spain) for endothelial MVs, anti-C235 PC7 (phycoerythrin-cyanine 7, Immunostep, Salamanca Spain) for erythrocyte MVs, anti-CD42a APC (allophycocyanin, Immunostep, Salamanca Spain) for platelet MVs and Annexin V–fluorescein isothiocyanate (FITC) (ImmunoTech SAS, Marseille, France). Megamix-Plus SSC beads (Biocytex, Marseille, France) calibrated from 0.16 to 0.5 mm were used to define an analysis window (gate) consistent with the size of MVs. Events less than 0.5 μm were identified in forward scatter and side scatter intensity dot representation, gated as a microparticle, and then plotted on 2-color fluorescence histograms. Double-positive events for Annexin and each antibody characterizing the MVs origin, in the MVs region were measured for endothelial, erythrocyte, and platelet MVs. MVs were then quantified using Flow Count Fluorospheres (Beckman Coulter): (MV counts × Fluorospheres concentration (counts/μL) × Fluorospheres volume (μL))/Fluorospheres counts × sample volume (μL). One independent flow cytometry specialist (E.Y.) that was not familiar with the participant’s clinical data performed the analysis using the FCS Express 4 (DeNovo Software, Glendale, CA, USA).

#### 4.2.4. Statistical Analysis

Transplant characteristics included in the analysis were: indication for transplant (disease, stage), phase at transplant, previous lines of treatment, conditioning (myeloablative versus reduced intensity and total body irradiation/ total body irradiation (TBI)-based versus non-TBI), type of donor (sibling, unrelated, haploidentical), acute GVHD grade II-IV and extensive chronic GVHD. Timing from transplant at study sampling and patient demographics were also included in the analysis. Sample size calculation was based on previous studies of platelet MVs in patients with increased CV risk compared to controls [12]. With a 5% level of significance and 80% power [50], the estimated sample size was 40 individuals per group.

Data were analyzed using the statistical program SPSS 23.0 (IBM SPSS Statistics for Windows, Version 23.0. Armonk, NY, USA: IBM Corp.). Descriptive statistics were performed using median and range for continuous variables and frequency for categorical variables. Continuous variables were assessed for normality and compared using one-way ANOVA with the Bonferroni correction or Kruskal–Wallis test. Follow-up was measured from the date of transplantation until the date of last follow up or death. Correlation between continuous variables was assessed with the Pearson’s or Spearman’s correlation coefficient. The level of statistical significance was defined at 0.05.

## 5. Conclusions

In conclusion, we demonstrate for the first time a crosstalk between endothelial injury and pro-coagulant activity in alloHCT survivors long after HCT using novel biomarkers, independently of traditional cardiovascular risk assessment. Identifying high-risk clinical features may guide the role of circulating MVs as novel markers of long-lasting subclinical pathophysiology.

## Figures and Tables

**Table 1 ijms-21-09768-t001:** Patients’ pre-transplant and transplant characteristics (*n* = 45).

Disease Type (n)	
AML	14
ALL	22
Lymphoma	4
MDS	2
Other	1
**Disease phase (n)**	
Early	28
Intermediate	12
Advanced	5
**Previous lines of treatment (median, range)**	4, 1–11
**Myeloablative conditioning (n)**	32
**TBI-based conditioning (n)**	12
**Donor**	
Sibling	19
Unrelated	30
Haploidentical	3
**Acute GVHD grade II-IV (n)**	20
**Extensive chronic GVHD (n)**	30

AML: acute myeloid leukemia; ALL: acute lymphoblastic leukemia; MDS: myelodysplastic syndrome; TBI: total body irradiation; GVHD: graft-versus-host disease.

**Table 2 ijms-21-09768-t002:** Baseline characteristics of cardiovascular disease in alloHCT survivors and matched controls.

	alloHCT (*n* = 45)	Controls (*n* = 45)	*p*-Value
**Age (years)**	46 ± 15	47 ± 9	0.214
**Office SBP (mmHg)**	121 ± 16	122 ± 14	0.770
**Office DBP (mmHg)**	80 ± 12	77 ± 12	0.324
**Heart rate (/min)**	78 (16)	75 (14)	0.125
**Total cholesterol (mg/dl)**	209 ± 26	199 ± 35	0.335
**LDL-C (mg/dl)**	140 ± 30	137 ± 25	0.267
**HDL-C (mg/dl)**	55 ± 10	48 ± 10	0.118
**Triglycerides (mg/dl)**	141 ± 51	127 ± 48	0.102
**Glucose (mg/dl)**	97 ± 10	88 ± 10	0.113
**BMI (kg/m^2^)**	28 (7)	28 (6)	0.605
**Smoking (%)**	4 (8.9)	7 (15.6)	0.824
Current	1 (2.2)	3 (6.6)	
Former	3 (6.6)	4 (8.8)	

alloHCT: allogeneic hematopoietic cell transplantation; SBP: systolic blood pressure; DBP: diastolic blood pressure; HDL-C: High-Density Lipoprotein Cholesterol, LDL-C: Low-Density Lipoprotein Cholesterol.

**Table 3 ijms-21-09768-t003:** Novel markers of endothelial dysfunction and thrombotic tendency in alloHCT survivors and matched controls.

	alloHCT (*n* = 45)	Controls (*n* = 45)	*p*-Value
**Platelet microvesicles (/μL)**	293 (432)	267 (205)	0.039
**Erythrocyte microvesicles (/μL)**	72 (156)	31 (431)	0.004
**Endothelial microvesicles (/μL)**	179 (54)	157 (85)	0.133

alloHCT: allogeneic hematopoietic cell transplantation. Continuous variables are presented as mean ± SD or median (interquartile range) according to normality tests.

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
