# Peer review of "Assessment of Endothelial Injury and Pro-Coagulant Activity Using Circulating Microvesicles in Survivors of Allogeneic Hematopoietic Cell Transplantation"

_ijms, 2020, doi:10.3390/ijms21249768_

Round 1
Reviewer 1 Report
In the study were analyzed circulating microvesicles in survivors of allogeneic hematopoietic cell transplantation (alloHCT) as markers of vascular damage and prothrombotic tendency.
I have some questions and comments:
1. How influenced the interval after alloHCT the erythrocyte and platelet MVs? It is also not clear how influenced the levels of these markers disease type.
2. At which time points were from patients collected blood samples used for MV quantitation?
3. Were collected blood samples and analyzed MV from the same patient before alloHCT and at different time points after alloHCT?
4. What was criterion for selection of control individuals?
5. In my opinion is for correct interpretation of obtained data limitation very heterogeneous group of patients after alloHCT (different disease type and phase, interval after alloHCT.
Author Response
In the study were analyzed circulating microvesicles in survivors of allogeneic hematopoietic cell transplantation (alloHCT) as markers of vascular damage and prothrombotic tendency. I have some questions and comments:
- How influenced the interval after alloHCT the erythrocyte and platelet MVs? It is also not clear how influenced the levels of these markers disease type.
- We thank the Reviewer for the kind comments. Timing from transplant and disease type were included in the analysis. We have updated the data in the revised manuscript. There was only a trend toward a decrease of endothelial MVs as the interval after alloHCT increases (p=0.094). There was no association for markers of disease.
- At which time points were from patients collected blood samples used for MV quantitation?
- During the study period, alloHCT survivors were referred from our HCT Clinic (January-December 2019) for performance of study procedures. According to study design, sampling was performed only at one time point for each patient.
- Were collected blood samples and analyzed MV from the same patient before alloHCT and at different time points after alloHCT?
- We understand the reviewer’s concern. As mentioned above, we did not plan for serial measurements. This has been added as a limitation of our study.
- What was criterion for selection of control individuals?
- As controls we enrolled consecutive individuals from our Outpatient Hypertension Clinic who attended regular appointments, matched for traditional cardiovascular risk factors (age, hypertension, diabetes, dyslipidemia, obesity, smoking) with alloHCT survivors.
- In my opinion is for correct interpretation of obtained data limitation very heterogeneous group of patients after alloHCT (different disease type and phase, interval after alloHCT.
- We agree with the reviewer and have added this limitation in our revised manuscript. Given the rarity of studies in the field, we believe that our study paves the way for future studies needed to clarify the effects of these parameters.
Reviewer 2 Report
Gavriilaki and colleaugues have performed an interesting study looking at microvesicles in adult post-HSCT survivors, and found raised erythrocyte and platelet MVs in HSCT recipients compared to controls. Endothelial MVs were raised but not significantly.
These are interesting observations, worthy of being published, but a more interesting control group would be patients with similar diseases who had not undergone HCT, or measurements in the same group of patients pre- and post-HCT.
Thus, I am not sure that the authors can conclude that HCT 'causes' the findings, as it may be the disease or pre-HCT treatment. I would suggest that the message is modified to reflect that.
Minor comments:
- The introduction could add some more information on the MV being associated with CV disease - for the non-expert reader, it is not clear why these markers were chosen.
2. At what point post-HCT were the measurements taken?
3. Lines 95 and 98 - the significant increases were found with respect to which group?
4. Line 100-101 - platelet MVs were significantly associated with endothelial (r=0.333, p=0.002) and erythrocyte (r=0.249, p=0.034) MVs. - clarify what you mean by this.
Author Response
Gavriilaki and colleaugues have performed an interesting study looking at microvesicles in adult post-HSCT survivors, and found raised erythrocyte and platelet MVs in HSCT recipients compared to controls. Endothelial MVs were raised but not significantly. These are interesting observations, worthy of being published, but a more interesting control group would be patients with similar diseases who had not undergone HCT, or measurements in the same group of patients pre- and post-HCT. Thus, I am not sure that the authors can conclude that HCT 'causes' the findings, as it may be the disease or pre-HCT treatment. I would suggest that the message is modified to reflect that.
- We are grateful to the reviewer for the kind comments. We understand the reviewer’s concern and include as one of our study’s limitations that no causal relationships can be assumed. To further highlight the reviewer’s comment we added relevant statements in the limitations and suggestions for future studies.
Minor comments:
- The introduction could add some more information on the MV being associated with CV disease - for the non-expert reader, it is not clear why these markers were chosen.
- We thank the reviewer for giving us the opportunity to add relevant information.
- At what point post-HCT were the measurements taken?
- Measurements were performed during the study visit, after a median of 2.3 (range 1.1-13.2) years. During the study period, alloHCT survivors were referred from our HCT Clinic (January-December 2019) for performance of study procedures. According to study design, sampling was performed only at one time point for each patient.
- Lines 95 and 98 - the significant increases were found with respect to which group?
The clarification has been added in the revised manuscript.
- Line 100-101 - platelet MVs were significantly associated with endothelial (r=0.333, p=0.002) and erythrocyte (r=0.249, p=0.034) MVs. - clarify what you mean by this.
- These three variables are expected to correlate with each other, as also shown our previous studies. This is only mentioned as a finding suggesting an interaction between MVs of different origins.
Round 2
Reviewer 1 Report
Authors answered questions and comments that I have addressed to them by revision of the previous version of manuscript and included some corresponding changes into revised manuscript.
Author Response
Thank you very much for your work in our manuscript.